# Poisoning the Search Space in Neural Architecture Search

**Robert Wu** [* 1]  **Nayan Saxena** [* 1]  **Rohan Jain** [* 1]

## Abstract

Deep learning has proven to be a highly effective problem-solving tool for object detection and image segmentation across various domains such as healthcare and autonomous driving. At the heart of this performance lies neural architecture design which relies heavily on domain knowledge and prior experience on the researchers' behalf. More recently, this process of finding the most optimal architectures, given an initial search space of possible operations, was automated by Neural Architecture Search (NAS). In this paper, we evaluate the robustness of one such algorithm known as Efficient NAS (ENAS) against data agnostic poisoning attacks on the original search space with carefully designed ineffective operations. By evaluating algorithm performance on the CIFAR-10 dataset, we empirically demonstrate how our novel search space poisoning (SSP) approach and multiple-instance poisoning attacks exploit design flaws in the ENAS controller to result in inflated prediction error rates for child networks. Our results provide insights into the challenges to surmount in using NAS for more adversarially robust architecture search.

## 1. Introduction

In the modern ecosystem, the problem of finding the most optimal deep learning architectures has been a major focus of the machine learning community. With applications ranging from speech recognition (Hinton et al., 2012) to image segmentation (Krizhevsky et al., 2012), deep learning has shown the potential to solve pressing issues in several domains including healthcare (Wang et al., 2016; Piccialli et al., 2021) and surveillance (Liu et al., 2016). However, a major challenge is to find the best architecture design for a given problem. This relies heavily on the researcher's

---
[*]Equal contribution [1]University of Toronto. Correspondence to: Nayan Saxena <nayan.saxena@mail.utoronto.ca>, Robert Wu <rupert.wu@mail.utoronto.ca>.

*Accepted by the ICML 2021 workshop on A Blessing in Disguise: The Prospects and Perils of Adversarial Machine Learning.* Copyright 2021 by the author(s).

domain knowledge and involves large amounts of trial and error. More recently, neural architecture search (NAS) algorithms have automated this dynamic process of creating and evaluating new architectures (Zoph & Le, 2016; Liu et al., 2018b;a). These algorithms continually sample operations from a predefined search space to construct architectures that best optimize a performance metric over time, eventually converging to the best child architectures. This intuitive idea greatly reduces human intervention by restricting human bias in architecture engineering to just the selection of the predefined search space (Elsken et al., 2019).

Although NAS has the potential to revolutionize architecture search across industry and research applications, human selection of the search space also presents an open security risk that needs to be evaluated before NAS can be deployed in security-critical domains. Due to the heavy dependence of NAS on the search space, poor search space selection either due to human error or by an adversary has the potential to severely impact the training dynamics of NAS. This can alter or completely reverse the predictive performance of even the most optimal final architectures derived from such a procedure. In this paper, we validate these concerns by evaluating the robustness of one such NAS algorithm known as Efficient NAS (ENAS) (Pham et al., 2018) against data-agnostic search space poisoning (SSP) attacks.

**Related Work**    A comprehensive overview of NAS algorithms can be found in Wistuba et al. (2019) and Elsken et al. (2019), with Chakraborty et al. (2018) summarising advances in adversarial machine learning including poisoning attacks. NAS algorithms have recently been employed in healthcare and applied in various clinical settings for diseases like COVID-19, cancer and cystic fibrosis (van der Schaar, 2020). Furthermore, architectures derived from NAS procedures have shown state of the art performance, often outperforming manually created networks in semantic segmentation (Chen et al., 2018), image classification (Real et al., 2019; Zoph et al., 2018) and object detection (Zoph et al., 2018). With rapid development of emerging NAS methods, recent work by Lindauer & Hutter (2020) has brought to light some pressing issues pertaining to the lack of rigorous empirical evaluation of existing approaches. Furthermore, while NAS has been studied to further develop more adversarially robust networks through addition of dense connections (Kotyan & Vargas, 2019; Guo et al.,

2020), little work has been done in the past to assess the adversarial robustness of NAS itself. Search phase analysis has shown that computationally efficient algorithms such as ENAS are worse at truly ranking child networks due to their reliance on weight sharing (Yu et al., 2019), which can be exploited in an adversarial context. Finally, most traditional poisoning attacks involve injecting mislabeled examples in the training data and have been executed against feature selection methods (Xiao et al., 2015), support vector machines (Biggio et al., 2012) and neural networks (Yang et al., 2017). To the authors' knowledge, no study, has approached poisoning in a data-agnostic manner, especially one that involves poisoning the search space in NAS. In summary, our main contributions through this paper are:

- We emphasize the conceptual significance of designing adversarial poisoning attacks that leverage the original search space and controller design in ENAS.
- We propose and develop the theory behind a novel data-agnostic poisoning technique called search space poisoning (SSP) alongside multiple-instance poisoning attacks, as described in Section 3.
- Through our experiments on the CIFAR-10 dataset in Section 4 we demonstrate how SSP results in child networks with inflated prediction error rates (up to $\sim 80\%$).

## 2. Background

### 2.1. Efficient Neural Architecture Search (ENAS)

**Search Space**    Consider the set $\mathcal{A}$ containing all possible neural network architectures or child models that can be generated. The ENAS search space is then represented as a directed acyclic graph (DAG) denoted by $\mathcal{G}$ which is the superposition of all child models in $\mathcal{A}$.

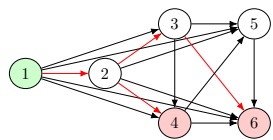

*Figure 1.* ENAS search space represented as a DAG. Red arrows represent one child model with input node 1 and outputs 4, 6 respectively.

Every node in Figure 2 represents local computations each having its own parameters with edges representing the flow of information between nodes. Sampled architectures are sub-graphs of $\mathcal{G}$ with parameters being shared amongst child models. Throughout this paper, we focus on the highly effective original ENAS search space as outlined in Pham et al. (2018) denoted by $\hat{\mathcal{S}}$ = {Identity, 3x3 Separable Convolution, 5x5 Separable Convolution, Max Pooling (3x3), Average Pooling (3x3)}.

**Search Strategy**    The ENAS controller is a predefined long short term memory (LSTM) cell which autoregressively samples decisions through softmax classifiers. The central goal of the controller is to search for optimal architectures by generating a child model $a \in \mathcal{G}$, feeding every decision on the previous step as an input embedding into the next step. Our main search strategy throughout this paper will be macro search where the controller makes two sampling decisions for every layer in the child network: (i) connections to previous nodes for skip connections, and (ii) operations to use from the search space.

**Performance Estimation**    As outlined in Pham et al. (2018), ENAS alternates between training the shared parameters $\omega$ of the child model $\mathbf{m}$ using stochastic gradient descent (SGD), and parameters $\theta$ of the LSTM controller using reinforcement learning (RL). First, keeping $\omega$ fixed, $\theta$ is trained with REINFORCE (Williams, 1992) and Adam optimizer (Kingma & Ba, 2014) to maximize the expected reward $\mathbb{E}_{\mathbf{m} \sim \pi(\mathbf{m};\theta)}[\mathcal{R}(\mathbf{m}, \omega)]$ (validation accuracy); and second, keeping the controller's policy $\pi(\mathbf{m}, \theta)$ fixed, $\omega$ is updated with SGD to minimize expected cross-entropy loss $\mathbb{E}_{\mathbf{m} \sim \pi}[\mathcal{L}(\mathbf{m}; \omega)]$. Note that different operations associated with the same node in $\mathcal{G}$ have their own unique parameters.

### 2.2. Training Data Poisoning

Traditionally, training data poisoning is defined as the adversarial contamination of the training set $T \subset \mathcal{D}$ by addition of an extraneous data point $(\mathbf{x}_p, \mathbf{y}_p)$ which maximizes prediction error across training and validation sets, while significantly impacting loss minimization during training (Xiao et al., 2015; Biggio et al., 2012; Muñoz-González et al., 2017; Yang et al., 2017). It is assumed here that the data is generated according to an underlying process $f : X \mapsto Y$, given a set $\mathcal{D} = \{\mathbf{x}_i, \mathbf{y}_i\}_{i=1}^{n}$ of *i.i.d* samples drawn from $p(X, Y)$, where $X$ and $Y$ are sets containing feature vectors and corresponding target labels respectively. While highly effective, existing poisoning techniques are highly data dependent and operate under the assumption that the attacker has access to training data. A more relaxed assumption would be to decouple the attack modality from training data and make it data agnostic, which is explored in the subsequent section.

## 3. Search Space Poisoning (SSP)

### 3.1. General Framework

Motivated by the previously described notion of training data poisoning, we introduce search space poisoning (SSP) focused on contaminating the original ENAS search space. The core idea behind SSP is to inject precisely designed ineffective operations into the ENAS search space to maximize the frequency of poor architectures appearing during

training. Our approach exploits the core functionality of the ENAS controller to sample child networks from a large computational graph of operations by introducing highly ineffective local operations into the search space. On the attacker's behalf, this requires no *a priori* knowledge of the problem domain or training dataset being used, making this new approach more favourable than traditional poisoning attacks. Formally, we describe a poisoned search space as $\mathcal{S} := \hat{\mathcal{S}} \cup \mathcal{P}$, where $\hat{\mathcal{S}}$ denotes the original ENAS search space operations and $\mathcal{P}$ denotes a non-empty set of poisonings where each poisoning is an ineffective operation. An overview of the SSP approach can be observed Figure 2.

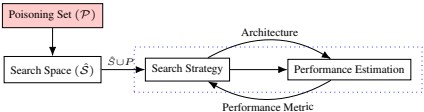

*Figure 2.* Overview of Search Space Poisoning (SSP)

### 3.2. Multiple-Instance Poisoning Attacks

Over the course of training, the LSTM controller paired with the RL search strategy in ENAS develops the ability to sample architectures with operations that most optimally reduce the validation error. As a result, single-instance poisoning attacks might not be as effective due to the tendency of the ENAS controller to draw fewer child networks with the single sub-optimal operation $o_\mathcal{P} \in \mathcal{P}$ over time. This behaviour of ENAS results in the algorithm almost entirely discarding networks with the singular $o_\mathcal{P}$ as training progresses. To circumvent this issue, we propose multiple-instance poisoning which essentially increases the likelihood of $o_\mathcal{P}$ being sampled from each poisoned search space. This is achieved by increasing the frequency of sampling $o_\mathcal{P}$ from $\mathcal{S}$ through inclusion of multiple-instances of each $o_\mathcal{P}$ from the poisoning multiset, so-called to allow for duplicate elements. An instance factor $q \in \mathbb{N}^{>0}$ would represent instance multiplication of $o_\mathcal{P}$ in the multiset $q$ times. Henceforth, the probabilities of sampling $o_{\hat{\mathcal{S}}} \in \hat{\mathcal{S}}$ and $o_\mathcal{P} \in \mathcal{P}$, respectively, are, $\Pr[o_{\hat{\mathcal{S}}}] := \frac{1}{|\mathcal{S}|+q|\mathcal{P}|} < \Pr[o_\mathcal{P}] := \frac{q}{|\mathcal{S}|+q|\mathcal{P}|}$. From this result it is evident that under a multiple-instance poisoning framework, the probability of sampling poisoned operations is higher as compared to operations in $\hat{\mathcal{S}}$.

### 3.3. Crafting Poisoning Sets with Operations

**Identity Operation**     The simplest way to attack the functionality of ENAS is to inject non-operations within the original search space which keep the input and outputs intact. As a result, the controller will sample child models with layers representing computations which preserve the inputs, making the operation highly ineffective within a network architecture. This goal is fulfilled by the identity operation which has no numerical effect on the inputs with

a minimal computational cost. It should also be noted that, the identity operation is not a skip connection. Therefore, we define our first set of poisonings as $\mathcal{P}_1 := \{\text{Identity}\}$.

**Transposed Convolutions**     As described earlier, amongst other useful operations the original ENAS search space $\hat{\mathcal{S}}$ also contains 3x3 and 5x5 convolutional layers (separable & non-seperable). With these settings under consideration, a more practical way of poisoning the search space is to reverse the effect of each of these convolutions. Given a normal convolutional layer $g$ and a transposed convolutional layer $h$ with the same parameters except for output channel sizes, $g$ and $h$ are approximate inverses. We achieve our goal of countering the effect of existing convolutions by including transposed convolutions in the set of poisonings resulting in our second poisoning set being $\mathcal{P}_2 := \{3\text{x}3 \text{ transposed convolution, 5x5 transposed convolution}\}$.

**Dropout Layer**     While dropout layers have historically been shown to be useful in preventing neural networks from over-fitting (Srivastava et al., 2014), a high dropout rate can result in severe information loss leading to poor performance of the overall network. This is because given a dropout probability $p \in [0, 1]$, dropout randomly zeroes out some values from the input to de-correlate neurons during training. We hypothesize that including such layers with high dropout probability, such as $p = 0.9$, has the potential to contaminate the search space with irreversible effects on the training dynamics of ENAS. Therefore, our final poisoning set is simply $\mathcal{P}_3 := \{\text{Dropout}(p = 0.9)\}$.

## 4. Experiments

*Table 1.* Summary of experimental search spaces.

| POISONING SET $\mathcal{P}_i$ | SEARCH SPACE $\mathcal{S}_i$ | EXPERIMENT | POISONING MULTISET $q(\mathcal{P}_i)$ |
|---|---|---|---|
| $\phi$ | $\hat{\mathcal{S}}$ | Original | $\varnothing$ |
| $\mathcal{P}_1 = \{\text{Identity}\}$ | $\mathcal{S}_1 = \hat{\mathcal{S}} \cup \mathcal{P}_1$ | 1a | $6(\mathcal{P}_1)$ |
| | | 1b | $36(\mathcal{P}_1)$ |
| | | 1c | $120(\mathcal{P}_1)$ |
| | | 1d | $300(\mathcal{P}_1)$ |
| $\mathcal{P}_2 = \{3\text{x}3 \text{ transposed convolution, }$ $5\text{x}5 \text{ transposed convolution}\}$ | $\mathcal{S}_2 = \hat{\mathcal{S}} \cup \mathcal{P}_2$ | 2a | $1(\mathcal{P}_2)$ |
| | | 2b | $6(\mathcal{P}_2)$ |
| | | 2c | $20(\mathcal{P}_2)$ |
| | | 2d | $50(\mathcal{P}_2)$ |
| $\mathcal{P}_3 = \{\text{Dropout}(p = 0.9)\}$ | $\mathcal{S}_3 = \hat{\mathcal{S}} \cup \mathcal{P}_3$ | 3a | $6(\mathcal{P}_3)$ |
| | | 3b | $36(\mathcal{P}_3)$ |
| | | 3c | $120(\mathcal{P}_3)$ |
| | | 3d | $300(\mathcal{P}_3)$ |
| $\mathcal{P}_4 := \mathcal{P}_1 \cup \mathcal{P}_2 \cup \mathcal{P}_3$ | $\mathcal{S}_4 = \hat{\mathcal{S}} \cup \mathcal{P}_4$ | 4a | $1(\mathcal{P}_4)$ |
| | | 4b | $6(\mathcal{P}_4)$ |
| | | 4c | $20(\mathcal{P}_4)$ |
| | | 4d | $50(\mathcal{P}_4)$ |

To test the effectiveness of our proposed approach, we designed experiments based on previously described methods as outlined in Table 1. Each experiment involved training ENAS on the CIFAR-10 dataset for 300 epochs on a cluster equipped with an Intel Xeon E5-2620 and Nvidia TITAN Xp GPU (hyperparameters used can be found in Appendix A). Code used to run our experiments can be found here. Across our experiments, errors increased consistently in re-

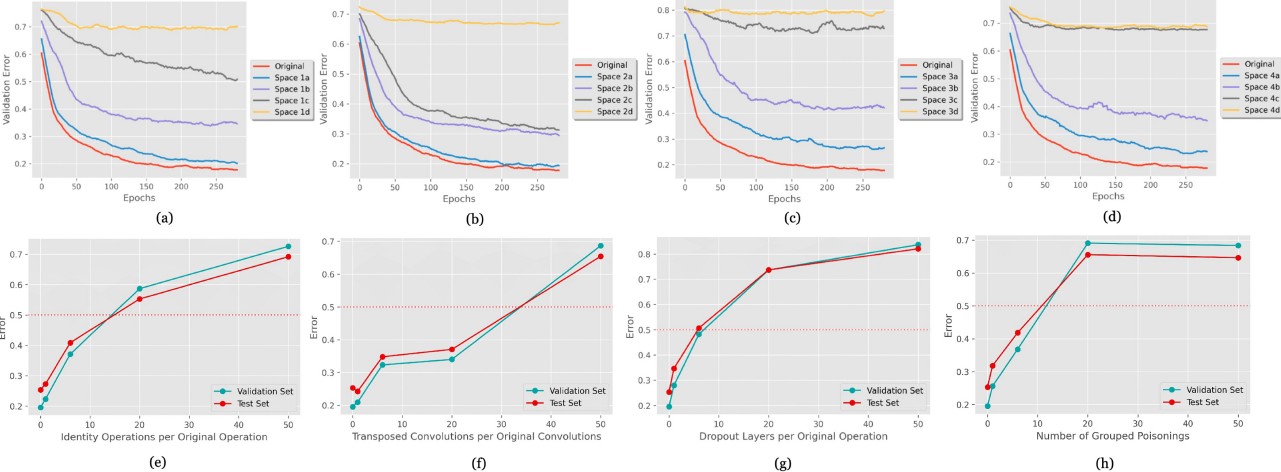

*Figure 3.* Experimental results for each search space outlined in Table 1. First row represents moving average of the validation error per 20 epochs for 300 total epochs, and second row represents final validation and test errors in classification as a function of multiple operation instances.

lation to the incremental addition of ineffective operations as visualised in Figure 3. A table of final validation and test errors can be found in Appendix B.

**Identity Operation**     Figure 3(a) shows that instance-multiplied identity operations increase the error considerably. Experiments 1b, 1c, 1d have several identity operations and resulted in high errors, with the extreme 69.19% in experiment 1d. In contrast, experiment 1a only has one identity per original operation and only raised error slightly to 27.28%. These results reinforce our hypothesis laid in Section 3.2. Figure 3(e) shows that excessive poisonings will result in diminishing returns.

**Transposed Convolutions**     Instance-multiplying transposed convolutions had mostly similar results of progressively increasing error as seen in Figure 3(b). We note that an instance factor of 50 (experiment 2d) results in an extreme increase in error at 68.83%; similar behaviour was observed in our other experiments but to a lesser degree. Figure 3(f) further shows that between the first four experiments, the increase in error slows down. However, the 100 transposed convolutions in experiment 2d show a staggering 28.36% increase in error.

**Dropout Layer**     Instance-multiplying dropout operations exhibited a similar pattern in validation to the previous operations, but the poisoning seemed to inflate the errors to a greater degree as seen in Figure 3(c). Figure 3(g) shows the experiments progressively worsening in error with experiment 3d hitting 83.69% validation and 82.07% test errors. We also observe that adding further dropout, like 300, results in smaller increases in error, like identity and unlike transposed convolutions operations. Dropout's pattern is similar to identity, but its effect on ENAS is more detrimental.

**Grouped Operations**     Graphing the validation error shows a sharper increase in error, implying that mixing different ineffective operations is more detrimental to ENAS than including several instances of the same operation. In reviewing Figures 3(d) and 3(h), we note that the 20 group poisonings in experiment 4c are about as effective as 300 identity or 100 transposed convolution operations (experiments 1d, 2d), and more effective than 36 dropout operations (experiment 3b). We also observe that experiments 4c, 4d were very close in training and final errors; the final errors were 65.55% and 64.64%, respectively. So by factor 20 in experiment 4c, we have reached the poisoning saturation point. In summary, grouping a variety of poisonings is more efficient than multiplying the same poisoning.

# 5. Conclusion

NAS algorithms present an important opportunity for researchers and industry leaders by enabling the automated creation of optimal architectures. However, it is also important to evaluate obvious vulnerabilities in these systems which can result in unforeseen model outcomes if not dealt with beforehand. Consistent with the earlier findings in Yu et al. (2019), our results highlight how the ENAS controller's dependence on parameter sharing leads to inaccurate predictions. We successfully demonstrated how using the same weights, although computationally cheap, compromises the functionality of ENAS when injected with poor operations. SSP successfully leveraged the inability of ENAS to alternate between weights shared across effective and ineffective operations as shown in our experimental results. These findings pave the way for machine learning researchers to explore improvements to the search space and controller design for more adversarially robust search.

## Acknowledgements

The authors would like to thank George-Alexandru Adam (Vector Institute; University of Toronto) for valuable comments and stimulating discussions that greatly influenced this paper. We are also grateful to Kanav Singla (University of Toronto) and Benjamin Zhuo (University of Toronto) for their initial contributions to the codebase.

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

# Appendix

## A. Hyperparameters

*Table 1.* Summary of experiment hyperparameters

| HYPERPARAMETER | VALUE |
| --- | --- |
| search_for | macro |
| batch_size | 128 |
| search_for | 300 |
| seed | 69 |
| cutout | 0 |
| fixed_arc | False |
| child_num_layers | 12 |
| child_out_filters | 36 |
| child_grad_bound | 5.0 |
| child_l2_reg | 0.00025 |
| child_keep_prob | 0.9 |
| child_lr_max | 0.05 |
| child_lr_min | 0.0005 |
| child_lr_T | 10 |
| controller_lstm_size | 64 |
| controller_lstm_num_layers | 1 |
| controller_entropy_weight | 0.0001 |
| controller_train_every | 1 |
| controller_num_aggregate | 20 |
| controller_train_steps | 50 |
| controller_lr | 0.001 |
| controller_tanh_constant | 1.5 |
| controller_op_tanh_reduce | 2.5 |
| controller_skip_target | 0.4 |
| controller_skip_weight | 0.8 |
| controller_bl_dec | 0.99 |
| p (Dropout Rate) | 0.9 |

## B. Supplementary Results

*Table 2.* Final validation and test errors across experiments.

| EXPERIMENT | VAL ERROR | TEST ERROR |
| --- | --- | --- |
| Original | 19.53 | 25.33 |
| 1a | 22.32 | 27.28 |
| 1b | 37.12 | 40.87 |
| 1c | 58.67 | 55.29 |
| 1d | **72.60** | **69**.19 |
| 2a | 20.95 | 24.25 |
| 2b | 32.33 | 34.78 |
| 2c | 33.99 | 37.05 |
| 2d | **68.63** | **65.41** |
| 3a | 27.94 | 34.68 |
| 3b | 48.17 | 50.61 |
| 3c | 73.63 | 73.70 |
| 3d | **83.69** | **82.07** |
| 4a | 25.60 | 31.81 |
| 4b | 36.80 | 41.81 |
| 4c | **69.05** | **65.55** |
| 4d | **68.35** | **64.64** |