# OpenReview forum: "Poisoning the Search Space in Neural Architecture Search"
_ICML.cc/2021/Workshop/AML — ICML 2021 Workshop AML Poster_

### Official Review · Reviewer_mZHE · 2021-06-19
**A NAS-targeted search space poison attack. Lack of comparing methods to valid its superiority.**

**Rating:** Accept
**Confidence:** 4

**Review:**

This paper shows the vulnerability of NAS algorithms, especially in the search space and controller design.

pros:
1. The paper is easy to follow.
2. The idea to poison the search space of NAS is interesting.
3. The pointed weakness of NAS algorithms is worthy to study.

cons:
1. The experiments are not strong. No comparison with former methods.
2. The proposed search space poison instances, Identity, Transposed Convolutions and Dropout Layer, are too limited.

---

### Decision · Program_Chairs · 2021-06-21

**Decision:**

Accept (Poster)

**Comment:**

This paper showed the vulnerability of NAS algorithms in the search space and controller design. The authors can further address the reviewer's comments.